# Ubiquitin-dependent folding of the Wnt signaling coreceptor LRP6

**Elsa Perrody[1†], Laurence Abrami[1†], Michal Feldman[1], Beatrice Kunz[1], Sylvie Urbé[2], F Gisou van der Goot[1]***

[1]Global Health Institute, Ecole Polytechnique Fédérale de Lausanne, Lausanne, Switzerland; [2]Institute of Translational Medicine, University of Liverpool, Liverpool, United Kingdom

**Abstract** Many membrane proteins fold inefficiently and require the help of enzymes and chaperones. Here we reveal a novel folding assistance system that operates on membrane proteins from the cytosolic side of the endoplasmic reticulum (ER). We show that folding of the Wnt signaling coreceptor LRP6 is promoted by ubiquitination of a specific lysine, retaining it in the ER while avoiding degradation. Subsequent ER exit requires removal of ubiquitin from this lysine by the deubiquitinating enzyme USP19. This ubiquitination-deubiquitination is conceptually reminiscent of the glucosylation-deglucosylation occurring in the ER lumen during the calnexin/ calreticulin folding cycle. To avoid infinite futile cycles, folded LRP6 molecules undergo palmitoylation and ER export, while unsuccessfully folded proteins are, with time, polyubiquitinated on other lysines and targeted to degradation. This ubiquitin-dependent folding system also controls the proteostasis of other membrane proteins as CFTR and anthrax toxin receptor 2, two poor folders involved in severe human diseases.

***For correspondence:** gisou. vandergoot@epfl.ch

[†]These authors contributed equally to this work

**Competing interests:** The authors declare that no competing interests exist.

## Introduction

While protein folding may be extremely efficient, the presence of multiple domains, in soluble or membrane proteins, greatly reduces the efficacy of the overall process. Thus, a set of enzymes and chaperones assist folding and ensure that a sufficient number of active molecules reach their final destination (*Brodsky and Skach, 2011*; *Ellgaard et al., 2016*). Even with help, folding may remain inefficient and thereby sensitive to errors or mutations, leading to disease. An illustrative example of folding inefficiency, and the consequences thereof, is the Cystic Fibrosis Transmembrane Regulator (CFTR), a chloride channel expressed at the surface of lung epithelial cells (*Riordan, 2008*). It has been estimated that less than 25% of newly synthesized CFTR molecules actually reach the plasma membrane. When the number of functional CFTR channels is further reduced, patients suffer from Cystic Fibrosis (*Riordan, 2008*). This is the case for mutations that affect the kinetics or thermodynamics of CFTR folding in the ER, such as the most frequent CF mutation ΔF508, and which are recognized by ER quality control systems and targeted for degradation (*Riordan, 2008*).

Inefficiency in folding is a fairly common characteristic of transmembrane proteins (*Abrami et al., 2008b*; *Deuquet et al., 2009*; *Guerriero and Brodsky, 2012*). Most transmembrane proteins, which jointly compose 30% of the human proteome, are synthesized by ER-bound ribosomes and co-translationally inserted into the ER membrane (*Ellgaard et al., 2016*; *Xu and Ng, 2015*). Subsequent folding must proceed in three topological environments: the ER lumen, the membrane and the cytosol. In each of these environments, the protein may benefit from the help of chaperones and folding enzymes. On the luminal side, these are fairly well characterized and include HSP40, HSP70 and HSP90 family members (*Brodsky and Skach, 2011*), the lectin chaperones calnexin and calreticulin (*Lamriben et al., 2016*) and protein disulfide isomerases (*Brodsky and Skach, 2011*). Less is known

**eLife digest** Proteins carry out almost every process that happens inside a cell. Like all machines, their ability to work properly depends on their three-dimensional shape and structure. To make proteins, building blocks called amino acids are first assembled into a string that, like wool in a sweater, needs to be knitted into the final three-dimensional structure. How proteins reach their 3D structure is called "folding", and when protein folding fails, or is not so efficient, it can cause very severe diseases.

Protein folding is not as nicely progressive as knitting a sweater: it is more like putting all the wool into a big messy blob that then suddenly turns into a protein with the right three-dimensional structure. Cells have machinery that can detect messy-looking molecules and destroy them. Therefore, new proteins need to be hidden from this machinery until they have finished folding.

A human protein called LRP6 is found on the surface of cells and it plays an important role in allowing cells to communicate with each other. Like many other proteins, LRP6 is produced inside the cell in a compartment called the endoplasmic reticulum and is then exported to the cell surface. In 2008, a team of researchers found that LRP6 is modified in a particular way known as S-palmitoylation before it leaves the endoplasmic reticulum. This suggested that there is a system that helps this protein to fold correctly.

Here Perrody, Abrami et al. – including some of the researchers from the previous work – used biochemical techniques to investigate how LRP6 folds. The experiments show that another type of protein modification that involves attaching a molecule called ubiquitin to LRP6 promotes this protein's folding. Once the protein is folded, the ubiquitin is removed from LRP6 by an enzyme called USP19. Further experiments show that this system also helps to ensure that two other important proteins fold correctly.

The next steps following on from this work are to identify the other molecules involved in this protein folding system. A future challenge is to find out how this system protects new proteins from being degraded while they are still folding.

about the assisted folding in the ER membrane and on the cytosolic side. If folding or assembly in any of these environments fails, the protein is recognized by quality control machineries and targeted to the ER associated degradation (ERAD) pathway (*Lemus and Goder, 2014*). ERAD involves polyubiquitination of the cytosolic domains of transmembrane protein, extraction from the ER membrane and degradation by the proteasome (*Lemus and Goder, 2014*).

Here we have studied the biogenesis of Low-density lipoprotein Receptor-related Protein 6 (LRP6), a key component of the canonical Wnt signaling pathway, which has been associated with many human pathologies including cancer, osteoporosis and metabolic diseases (*Joiner et al., 2013*), but also involved in the formation of gap junctions in cardiomyocytes (*Li et al., 2016*). LRP6 is a type I membrane protein composed of a large extracellular domain containing multiple ß-propeller and EGF-like domains (*MacDonald et al., 2009*). Proper folding of the ß-propeller domains depends of the dedicated LRP6 chaperone Mesd (mesoderm development) (*Hsieh et al., 2003*) and possibly other chaperones such as calnexin. We have previously shown that exit of LRP6 from the ER also requires a cytoplasmic post-translational modification, namely S-palmitoylation, on two cysteine residues in close proximity to the transmembrane domain (*Abrami et al., 2008b*). LRP6 acylation influences the conformation of this domain, possibly by alleviating the hydrophobic mismatch between the thickness of the membrane and the length of the hydrophobic stretch composing the transmembrane domain. Mutation of the palmitoylation sites leads to retention of LRP6 in the ER and ubiquitination on Lys-1403 (*Abrami et al., 2008b*) indicative of its recognition by a quality control mechanism (*Feldman and van der Goot, 2009*). Once properly folded, LRP6 exits the ER, it transits through the Golgi, as indicated by the acquisition of Endoglycosidase H insensitive complex sugars (*Abrami et al., 2008b*), before reaching the plasma membrane where its signal role takes place.

Here we have further investigated LRP6 biogenesis in the ER. We show that newly synthesized wild type LRP6 also undergoes ubiquitination, on the same site (Lys-1403), and that this post-

translational modification promotes its folding. For ER exit to occur, ubiquitination must however be subsequently removed. This is mediated by the deubiquitinating enzyme USP19, one of the isoforms of which is tail-anchored to the ER (*Hassink et al., 2009*; *Wing, 2016*). It is thought to play an essential role in skeletal muscle atrophy through unclear mechanisms (*Wing, 2016*). At the more mechanistic level, USP19 has been shown to control the stability of several cytoplasmic proteins such as the inhibitors of apoptosis c-IAP1, and c-IAP2 (*Mei et al., 2011*), hypoxia-inducible factor 1-alpha (HIF-1α) (*Altun et al., 2012*) and the initiator of autophagy Beclin1 (*Jin et al., 2016*). Interestingly, USP19 is a target gene of the unfolded protein response (UPR) and was found to rescue CFTR ΔF508 from ERAD (*Hassink et al., 2009*). More recently it was found that USP19 is a key player in a novel misfolded-protein associated secretion MAPS pathway (*Lee et al., 2016*). We show that USP19 controls the cellular levels of LRP6. In the absence of USP19, LRP6 is efficiently targeted to ERAD. Upon USP19 overexpression, ER exit of LRP6 is greatly enhanced.

This work reveals the existence of a novel assisted-folding system that operates on the cytosolic side of the ER and depends on site-specific cycles of ubiquitination-deubiquitination. Upon ubiquitination on Lys-1403, LRP6 presumably interacts with an ubiquitin-binding protein that acts as a chaperone and provides the protein with time to fold. If folding is successful, LRP6 is transported out of the ER, a step that is favored by palmitoylation (*Abrami et al., 2008b*). If folding is impaired or delayed, polyubiquitination of other cytosolic lysine residues takes over and targets LRP6 to ERAD. The here identified ubiquitination-deubiquitination assisted-folding cycle also appears to operate on other membrane proteins such as CFTR or the anthrax toxin receptor 2 (also know as Capillary Morphogenesis gene 2, CMG2), loss of function of which leads to Hyaline Fibromatosis Syndrome (*Deuquet et al., 2012*).

## Results and discussion

### Massive degradation of newly synthesized LRP6 in the ER

We analyzed LRP6 stability by performing a cycloheximide chase, which consists in following total protein expression levels by western blot analysis of extracts from Pigmented epithelial (RPE1) cells submitted for various times to a protein synthesis block. No significant decay of LRP6 was observed over a 6 hr period (*Figure 1A,B*). We next measured LRP6 stability by performing $^{35}$S Cys/Met metabolic pulse-chase experiments. Protein decay, monitored following a 20 min metabolic pulse, indicated that the apparent half-life ($t_{\frac{1}{2}}^{app}$ of endogenous LRP6 in RPE1 cells is approximately 3 hr (*Figure 1C,D*, and supplementary information in [*Abrami et al., 2008b*]). The same experiment was repeated on transiently expressed myc-LRP6 in HeLa cells (*Figure 1—figure supplement 1*). Then, we show that $t_{\frac{1}{2}}^{app}$ was identical for endogenous LRP6 in RPE1 cells and transiently expressed myc-LRP6 in Hela, indicating that the potential difference in expression in these two systems does not affect degradation rates and allowing us to use both systems.

The above apparent discrepancy between the cycloheximide chase and the metabolic labeling approach is due to the fact that stability of mature LRP6 is monitored through the first approach, while newly synthesized LRP6 is monitored by the second. Considering the events that occur following synthesis of a membrane protein –more or less efficient folding, ER exit, transport to destination–, the $t_{\frac{1}{2}}^{app}$ of a protein determined using metabolic pulse-chase experiments may greatly dependent on the duration of the pulse. Indeed, the $t_{\frac{1}{2}}^{app}$ of LRP6 increased to ≈ 5 hr for a 2 hr pulse and >15 hr for a 16 hr pulse (*Figure 1C,D,E*). The long apparent half-lives estimated both by long metabolic labeling and cycloheximide chase suggest that mature LRP6 has a half-life that exceeds 20 hr. The rapid decay times observed for short metabolic pulses on the other hand indicate that close to 80% of newly synthesized LRP6 molecules are degraded in our cells within the first 6 hr (*Figure 1D*), and thus do not contribute to the population of active Wnt signaling co-receptors.

Cellular degradation of membrane proteins can occur by two main pathways: ERAD, where proteolysis is mediated by the proteasome, or the lysosomal pathway, which is responsible for degradation of most membrane proteins but is also the endpoint of autophagy (*Figure 1I*). To determine the relative contribution of these pathways to the degradation of LRP6, we repeated the 20 min metabolic pulse-chase experiments while either inhibiting the proteasome with MG132 or preventing lysosomal degradation by inhibiting the vacuolar ATPase with Bafilomycin A. MG132 protected LRP6

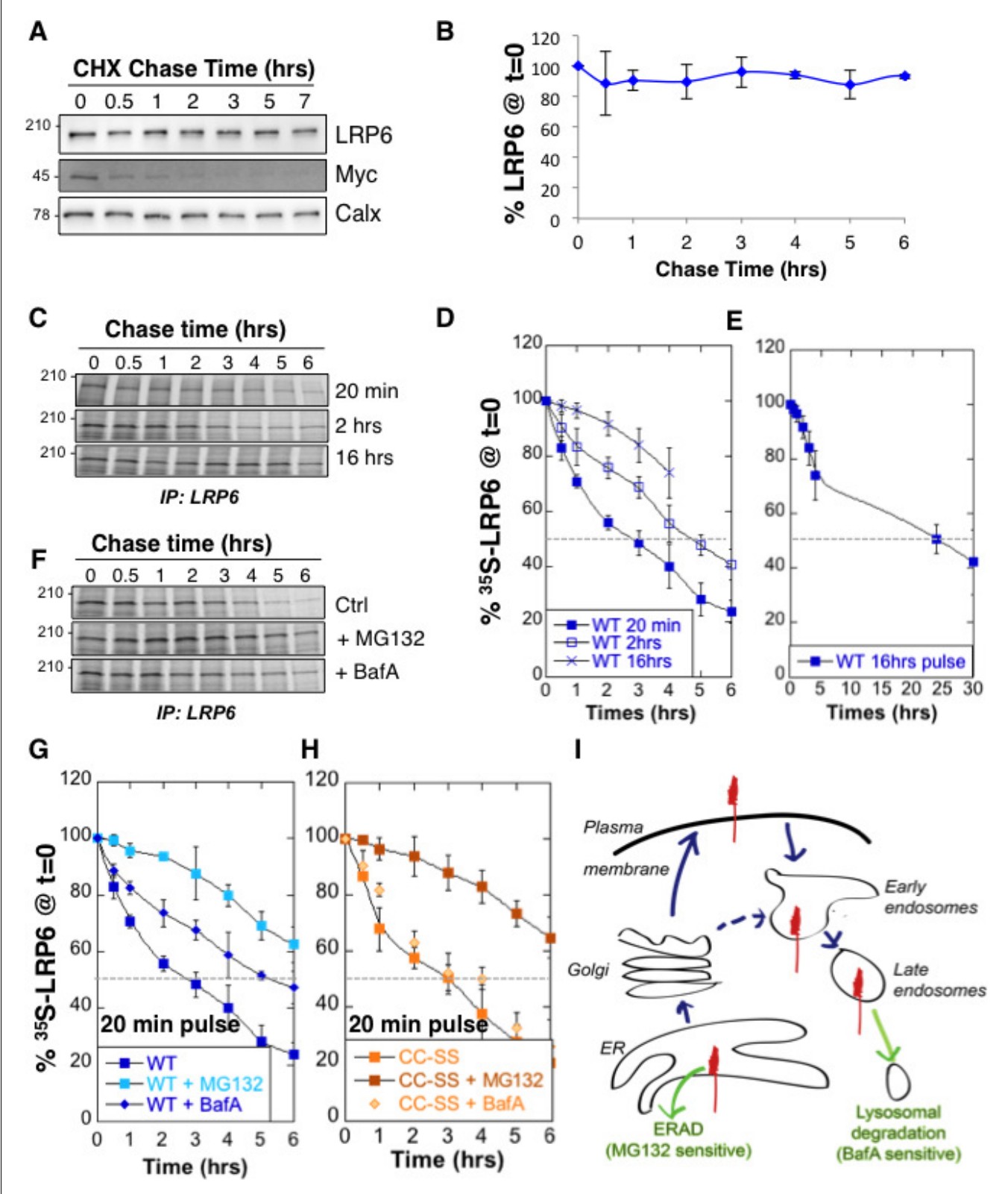

**Figure 1.** LRP6 undergoes rapid degradation following synthesis in the ER but is stable once mature. (A) representative western blot of a cycloheximide (CHX) chase in RPE1 cells. 40 μg of total cell extracts from RPE1 cells were loaded per lane, analyzed by SDS-PAGE followed by Western blotting against endogenous LRP6, calnexin (Calx), a stable protein and Myc, a short lived protein. (B) Experiments as in A were quantified by ImageJ software, n = 3. **CDE**: RPE1 cells were submitted to metabolic $^{35}$S Cys/Met labeling for different times and subsequently chased for different times. Endogenous

*Figure 1 continued on next page*

*Figure 1 continued*

LRP6 was immunoprecipitated with an anti-LRP6 antibody. A representative experiment in shown in (**C**). Autoradiograms were quantified using the Typhoon imager and means of different experiments were calculated (**D** and **E**). Error bars represent standard deviation (n = 6 for the 20 min pulse; n = 4 for the 2 hr and 16 hr pulses). **FG**: RPE1 cells were treated or not with MG132 or Bafilomycin A and subsequently submitted, in the presence or not of the drugs, to metabolic a 20 min $^{35}$S-Cys/Met pulse followed by different chase times. A representative experiment in shown in (**F**). Errors represent standard deviation (n = 4 for MG132; n = 3 for Bafilomycin A, BafA, the WT control curve corresponds to that shown in *Figure 1D*). (**G**) Hela cells transiently expressing myc-tagged palmitoylation deficient LRP6 (CC–SS) were submitted to metabolic $^{35}$S Cys/Met labeling for different times and subsequently chased for different times. LRP6 was subsequently immunoprecipitated using an anti-myc antibody. Errors represent standard deviation (n = 3). **H**: Cartoon depicting the two major cellular degradation pathways for membrane proteins: ERAD (blocked by MG132) and lysosomal pathway (blocked by Bafilomycin A).

The following source data and figure supplements are available for figure 1:

**Source data 1.** Numeric data for graphs of *Figure 1B,D,E and F* and 1 hr.

**Figure supplement 1.** Variation in USP19 cellular amount influences LRP6 degradation rates.

**Figure supplement 1—source data 1.** Numeric data for graphs of *Figure 1—figure supplement 1A*.

from degradation early after synthesis (*Figure 1F,G*), consistent with degradation of a subpopulation of LRP6 molecules by ERAD during biogenesis. Bafilomycin A protected LRP6 at later times (*Figure 1F,G*), consistent with transport of a portion of newly synthesized LRP6 molecules to the plasma membrane and their subsequent endocytosis and targeting to lysosomes (*Figure 1I*). That LRP6 undergoes degradation by two distinct pathways is further support by the observation that palmitoylation deficient LRP6, in which the two cysteines are mutated to serines (LRP6$^{CC-SS}$), and which does not exit the ER (*Abrami et al., 2008b*), is partially rescued by MG132 (*Figure 1H*) but is insensitive to inhibitors of lysosomal enzymes such as Bafilomycin A (*Figure 1H*) or leupeptin (supplementary information in [*Abrami et al., 2008b*]). The involvement of lysosomal enzymes in LRP6 degradation thus requires export out of the ER.

## Role of palmitoylation and ubiquitination in LRP6 biogenesis

We have previously shown that LRP6 undergoes palmitoylation in the ER on Cys-1394 and Cys-1399 and that palmitoylation is required for ER exit (*Abrami et al., 2008b*). This was based on the observation that palmitoylation deficient LRP6 did not exit the ER and underwent ubiquitination on Lys-1403 (*Abrami et al., 2008b*). To further investigate the importance of palmitoylation and ubiquitination on LRP6 biogenesis, we performed $^{35}$S Cys/Met metabolic pulse-chase experiments on the palmitoylation deficient LRP6$^{CC-SS}$ mutant, on the K1403R mutant (LRP6$^{CC-SS}$) and on the triple mutant (LRP6$^{CC-SS}_{K1403R}$). Following a 20 min pulse, no significant difference in degradation was observed between wild type LRP6 and LRP6$^{CC-SS}$ (*Figure 2A*) in agreement with our previous observations (*Abrami et al., 2008b*). In contrast, degradation was accelerated by the K1403R mutation (*Figure 2A*).

To estimate the importance of the contribution of Lys-1403 to LRP6 folding, we compared the effect of this mutation to that of silencing the dedicated LRP6 chaperone Mesd. Remarkably, mutation of Lys-1403 had a more pronounced effect on LRP6 degradation than *mesd* silencing (*Figure 2B*). Accelerated LRP6$_{K1403R}$ degradation did not involve lysosomes since Bafilomycin A had no effect, also suggesting that newly synthesized LRP6$_{K1403R}$ does not significantly exit the ER during the 6 hr that follow its synthesis. Degradation of LRP6$_{K1403R}$ could however be partially rescued by MG132 (*Figure 2C*).

Consistent with its targeting to the proteasome, LRP6$_{K1403R}$ underwent polyubiquitination as revealed when immunoprecipitating LRP6 from MG132 treated cells and blotting against ubiquitin (*Figure 2D*). This observation also shows that ERAD targeting of LRP6 does not involve, or at least does not require, Lys-1403 polyubiquitination.

Extending the pulse time to 2 or 16 hr revealed that the stability of LRP6 does depend on palmitoylation (*Figure 2E,F*). The triple LRP6$^{CC-SS}_{K1403R}$ mutant was therefore the least stable (*Figure 2F*).

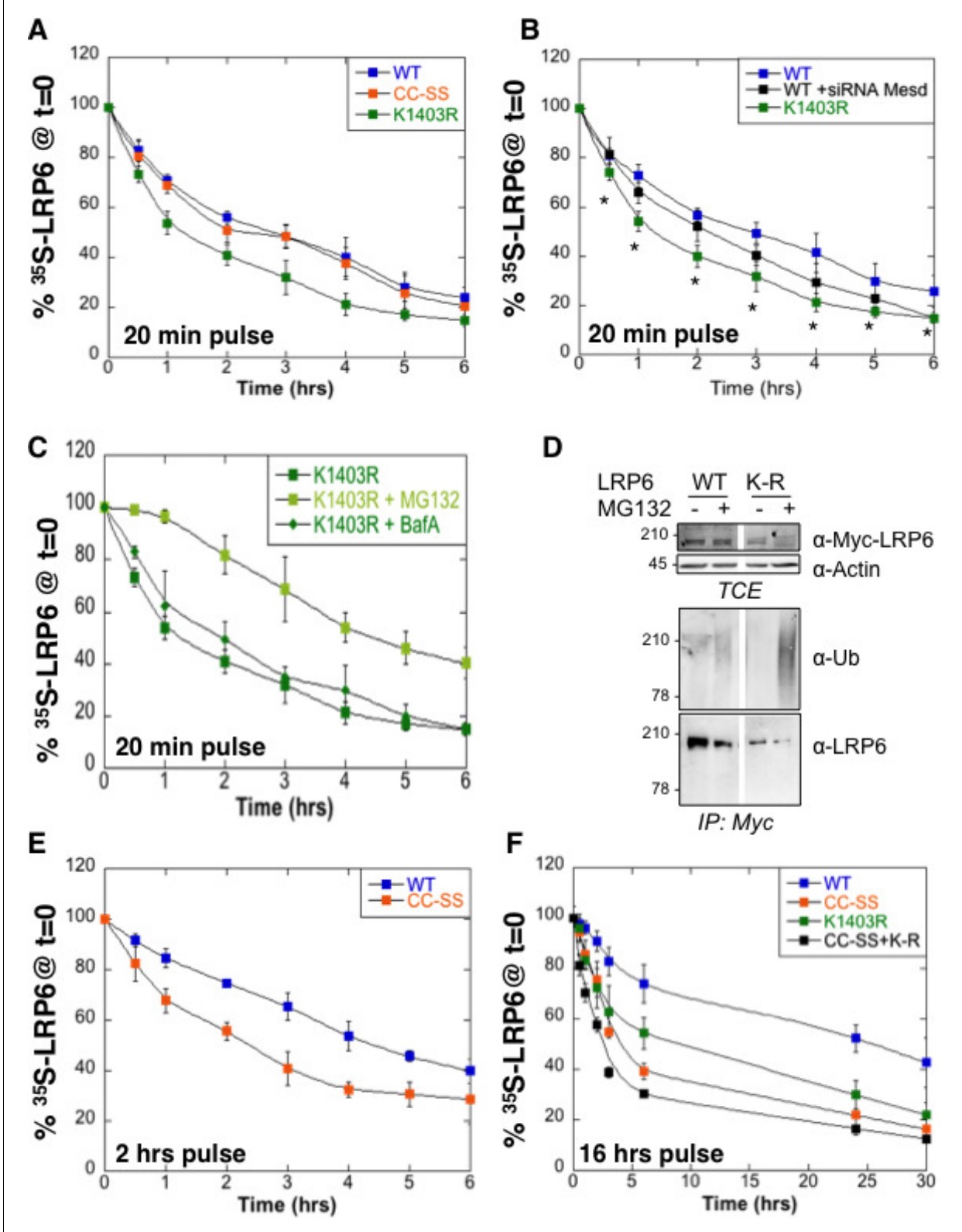

**Figure 2.** Mutation of the palmitoylation sites and the Lys-1403 ubiquitination site accelerate LRP6 targeting to ERAD. (**A**) Metabolic $^{35}$SCys/Met pulse chase experiment (20 min pulse) on transiently expressed myc-LRP6 wild type (WT, curve corresponding to the one in *Figure 1D*), palmitoylation deficient (CC-SS, n = 3) or K1403R (KR, n = 6) mutants in HeLa cells. (**B**) Metabolic $^{35}$SCys/Met pulse chase experiment (20 min pulse) on transiently expressed myc-LRP6 wild type (WT, n = 7), or K1403R (K1403R, n = 7) mutant in HeLa cells silenced or not for *mesd* gene (siRNA mesd, n = 3). Errors
*Figure 2 continued on next page*

*Figure 2 continued*

represent standard deviation,*<p=0.05 calculated between LRP6 WT and K1403R. (**C**) Metabolic $^{35}$SCys/Met pulse chase experiment (20 min pulse) on transiently expressed myc-LRP6K1403R in HeLa cells supplemented or not (n = 6, curve corresponding to the one in **Figure 2A**) with MG132 (K1403 + MG123, n = 3) or Bafilomycin A (K1403 + BafA, n = 3). (**D**) Immunoprecipitation of myc-tagged LRP6 Wild Type (WT) and K1403 mutant (KR) revealed with anti-Ubiquitin antibody, with or without MG132 treatment. (**E**) Metabolic $^{35}$SCys/Met pulse chase experiment (2 hr pulse) on transiently expressed myc-LRP6 wild type (WT, n = 4, curve corresponding to the one in **Figure 1D**) or palmitoylation deficient (CC-SS, n = 3) in HeLa cells. (**F**) Metabolic $^{35}$SCys/Met pulse chase experiment (16 hr pulse) on transiently expressed myc-LRP6 wild type (WT, curve corresponding to the one in **Figure 1D**), palmitoylation deficient (CC-SS, n = 3), K1403R (K1403R, n = 3) or K1403R in the palmitoyl deficient background (CC-SS + KR, n = 3) mutants in HeLa cells.

The following source data is available for figure 2:

**Source data 1.** Numeric data for graphs of *Figure 2A,B,C and E,F*.

Altogether these observations indicate that spontaneous folding of LRP6 is very inefficient, and that both ubiquitination and palmitoylation promote LRP6 biogenesis and ER exit. The process is however not all-or-none, *i.e.* even in the absence of Lys-1403 and/or palmitoylation, a small population of molecules folds properly and exits the ER. Pulse-chase experiments with long $^{35}$S pulses indeed reveal biphasic decay curves for all mutants studied, and show the existence, irrespective of the mutations, of a minor population of extremely long-lived molecules, which presumably reside at the plasma membrane. Importantly these constitute the steady state population that is revealed by western blotting. Western blot analysis of LRP6 mutants may therefore be misleading as to the importance of specific residues for biogenesis and membrane targeting (*Abrami et al., 2008b*).

## Working hypothesis

The above findings, combined with our previous observation that LRP6$^{CC–SS}$ undergoes ubiquitination on Lys-1403 and fails to exit the ER (*Abrami et al., 2008b*), led us to propose the following working hypothesis: following synthesis and insertion into the ER membrane, LRP6 first undergoes ubiquitination, probably of a specific type, on Lys-1403, allowing it to interact with an ER or cytosolic ubiquitin-binding protein. This interaction provides LRP6 with time to fold –and as such this ubiquitin-binding protein would act as a chaperone– protecting it from ERAD targeting. Lys-1403 is then deubiquitinated. At this stage, LRP6 can either (1) undergo palmitoylation of its two juxtamembranous cysteine residues, followed by ER exit, (2) be re-ubiquitinated on Lys-1403, or (3) be polyubiquitinated on one of the other 16 cytoplasmic lysine residues and sent to ERAD. Option 1, re-ubiquitination on Lys-1403, allows LRP6 to undergo a second cycle of interaction with its ubiquitin-binding chaperone, further promoting folding. Palmitoylation, which favors ER exit (*Abrami et al., 2008b*), could prevent LRP6 from Lys-1403 ubiquitination, possibly through a conformational change of steric hindrance given the juxtamembrane localization of both modifications.

A major prediction of this model is the existence of an ER localized deubiquitinating (DUB) enzyme that can remove ubiquitin specifically from Lys-1403. This DUB would control LRP6 biogenesis.

## USP19 promotes LRP6 biogenesis and controls Wnt signaling

The human genome encodes some 100 DUBs (*Clague et al., 2013*). Of these, the ubiquitin-specific proteases (USPs) represent the largest family with 56 members (*Ye, 2006*). Because LRP6 folding takes place in the ER, we searched for ER-associated DUBs and to the best of our knowledge, only USP19 has an ER localized isoform (*Clague et al., 2013*; *Hassink et al., 2009*; *Wing, 2016*). Endogenous LRP6 and USP19 could be co-immunoprecipitated from RPE1 cells (*Figure 3A*). While ubiquitination of LRP6 is undetectable under control condition (*Figure 3A*; see also *Abrami et al., 2008b*), silencing of *usp19* (*Figure 3—figure supplement 1A*) revealed a clear LRP6 ubiquitination signal (*Figure 3A*), often, but not always, appearing as a well identifiable band and a smear. Strikingly, LRP6 levels dropped drastically upon *usp19* silencing whether in HeLa (*Figure 3—figure supplement 1B*), RPE1 cells or primary human fibroblasts (*Figure 3B* and *Figure 3—figure supplement 1C*). The same was observed upon CRISPR/Cas9-mediated *usp19* knockout in RPE1 cells (*Figure 1—figure supplement 1B*). Silencing of *usp19* also led to an increase in ubiquitination of

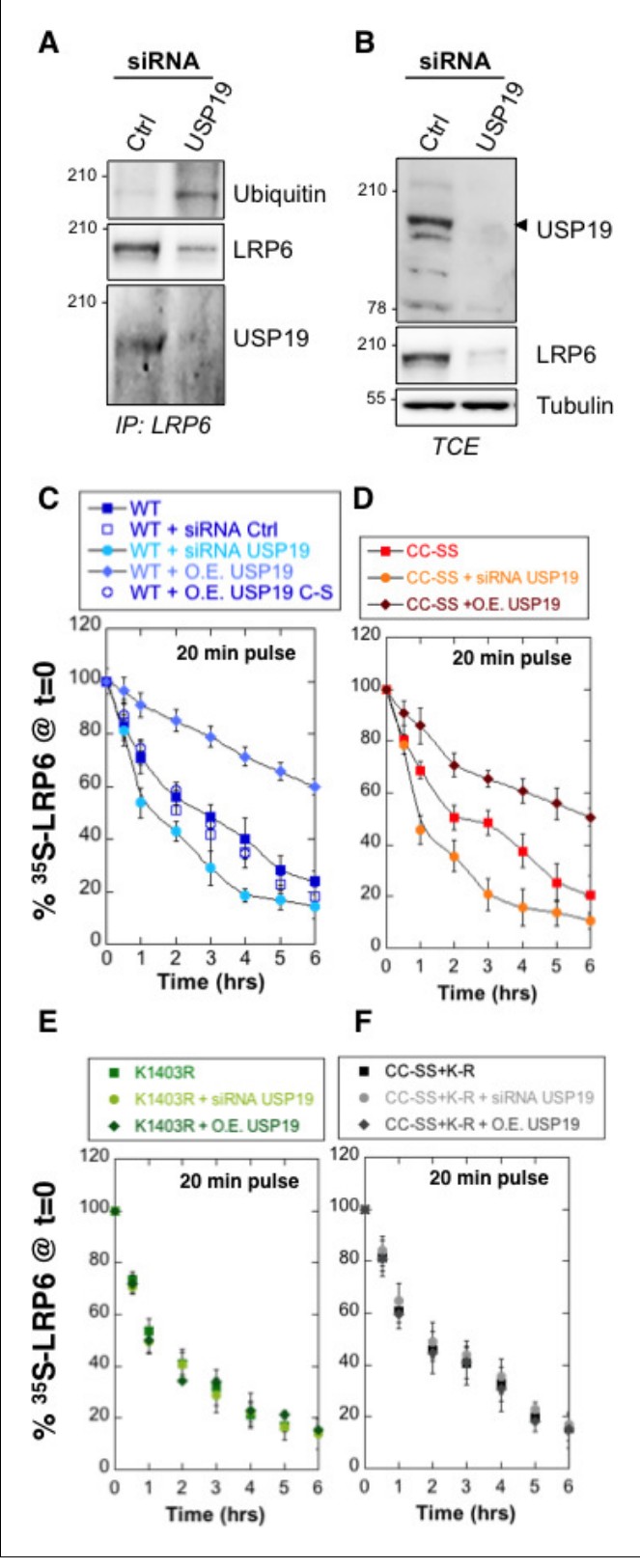

**Figure 3.** De-ubiquitination of LRP6 by USP19 on Lys-1403 promotes LRP6 biogenesis. (**A**) Immunoprecipitation of endogenous LRP6 and (**B**) cellular level in RPE1 cells upon *usp19* silencing. C/D/E/F: Metabolic $^{35}$SCys/Met pulse chase experiment (20 min pulse) on transiently expressed myc-LRP6 wild type (WT), palmitoylation deficient mutant (CC–SS), K1403R mutant (K1403R) or K1403R mutation in the palmitoyl deficient background (CC-SS + KR)

*Figure 3 continued on next page*

*Figure 3 continued*

mutants in HeLa cells upon over expression of GFP-tagged USP19 (O.E. USP19) or GFP-tagged USP19 catalytically inactive (O.E. USP19 C-S) or upon *usp19* gene silencing (siRNA USP19). (C) WT n = 6, same as in *Figure 1D*, other conditions n = 3, (D) (CC-SS curve corresponding to the one in *Figure 2A*) and (F) all conditions n = 3, (E) K1403R n = 6, other conditions n = 3.

The following source data and figure supplements are available for figure 3:

**Source data 1.** Numeric data for graphs of *Figure 3C, D and E, F*.
**Figure supplement 1.** Variation in USP19 cellular amount influences LRP6 ubiquitination state.
**Figure supplement 1–source data 1.** Numeric data for graphs of *Figure 3—figure supplement 1A, C* .

palmitoylation-deficient LRP6$^{CC–SS}$ and a decrease of its expression (*Figure 3—figure supplement 1D*). In reverse, over expression of wild type USP19, but not of its catalytically inactive mutant (USP19$^{C506S}$ [*Hassink et al., 2009*]) reduced ubiquitination of LRP6$^{CC–SS}$ (*Figure 3—figure supplement 1E*), indicating that USP19 influences LRP6 ubiquitination in a manner that depends on its DUB activity. Thus consistent with our working hypothesis, an ER-localized DUB, USP19, controls the expression of LRP6.

We next investigated whether USP19 indeed affects LRP6 biogenesis. siRNA-mediated silencing of *usp19* in HeLa cells and *usp19* knock-out in RPE1 cells (*Figure 1—figure supplement 1B*) both led to a strong decrease in $t_{\frac{1}{2}}^{app}$ of LRP6, droping from 3 to 1 hr, as monitored by $^{35}$S Cys/Met metabolic pulse-chase experiments (*Figure 3C* and *Figure 1—figure supplement 1A*). Overexpression of wild type USP19 led to a marked increase in $t_{\frac{1}{2}}^{app}$ to above 6 hr, beyond the time frame of these experiments. Importantly, overexpression of catalytically inactive USP19$^{C506S}$ had no effect. This not only shows that the catalytic activity of USP19 is required, but the complete absence of rescue also indicates that USP19 does not act as a chaperone, as proposed for the unconventional misfolded protein-associated secretion MAPS pathway (*Lee et al., 2016*).

The palmitoylation deficient LRP6$^{CC–SS}$ was affected in a qualitatively similar manner: silencing *usp19* decreased $t_{\frac{1}{2}}^{app}$ to less than 1 hr, while over expression of USP19 increased $t_{\frac{1}{2}}^{app}$ to 6 hr (*Figure 3D*). Remarkably, silencing or overexpression of USP19 had no effect whatsoever on LRP6$_{K1403R}$ (*Figure 3E*). Mutation of Lys-1403 similarly abolished the regulatory effect of USP19 on LRP6$^{CC–SS}$, $LRP6_{K1403R}^{CC–SS}$ being insensitive to the cellular USP19 levels (*Figure 3F*). Altogether these experiments clearly point to Lys-1403 as the specific target site of USP19, revealing the crucial role of this residue in preventing early LRP6 degradation. These experiments also show that USP19, even when overexpressed, cannot reverse the polyubiquination undergone by $LRP6_{K1403R}$ and thus ERAD targeting, further highlighting the specificity of this DUB. Finally, these observations demonstrate that USP19 promotes the biogenesis of LRP6, with only 18% of newly synthesized LRP6 molecules surviving beyond 6 hr in the absence of USP19 and 60% surviving upon USP19 overexpression.

We finally tested the importance of USP19 for LRP6 function, as the co-receptor in Wnt signaling. Overexpression of USP19 in HeLa cells lead to a 50% increase in the Wnt signaling capacity (*Figure 4A*), monitored using the TOPFLASH reporter assay (*Abrami et al., 2008a*). Most strikingly, silencing of usp19, but not usp13, in RPE1 cells stably expressing 7xTCF-FFluc directly activated by the TCF/-βcatenin complex led to a more than 80% drop in Wnt signaling (*Figure 4B*). In agreement, surface biotinylation showed that *usp19* silencing lead to a drastic drop LRP6 at the cell surface (*Figure 4—figure supplement 1*).

## Concluding remarks

We here reveal the existence of an ubiquitin-dependent folding machinery, which operates on the cytosolic side of the ER membrane (*Figure 4C*). More specifically we show that following synthesis, LRP6 undergoes ubiquitination on Lys-1403, most likely of a specific form. In conceptual analogy to the binding of newly synthesized mono-glucosylated proteins to calnexin in the ER lumen (*Lamriben et al., 2016*), we hypothesize that Lys-1403-ubiquitin can interact with a putative ER-

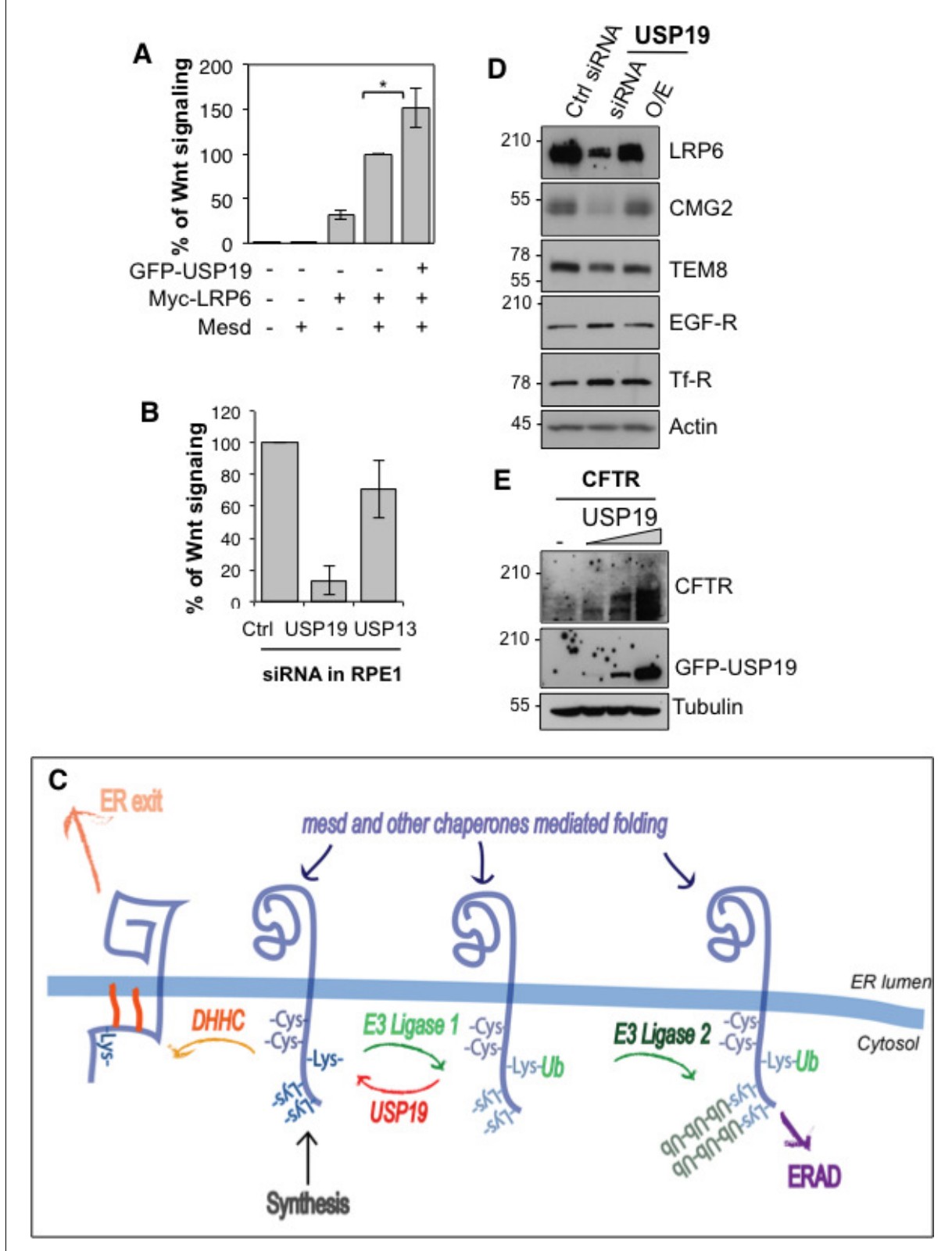

**Figure 4.** USP19 controls the Wnt signaling capacity of the cell. (**A**) Wnt signaling measured in HEK293 cells carrying the TOPFLASH reporter assay, transiently transfected for the indicated constructs (n = 5) (**B**) Wnt signaling measured in RPE1 cells stably expressing lentiviral vector possessing a 7xTCF-FFluc upon *usp19* or *usp13* silencing (n = 4). (**C**) Working model (described in the text). (**D**) cellular level of the indicated endogenous proteins in
*Figure 4 continued on next page*

*Figure 4 continued*

RPE1 cells upon *usp19* silencing (siRNA) or overexpression (O.E.) of GFP-tagged USP19. (E) cellular level of transiently co-transfected CFTR (constant amount) and GFP-tagged USP19 (increasing amount) in RPE1 cells.

The following source data and figure supplements are available for figure 4:

**Source data 1.** Numeric data for graphs of *Figure 4A, B*.

**Figure supplement 1.** *usp19* silencing leads to decrease in LRP6 cell surface expression.

**Figure supplement 1–source data 1.** Numeric data for graphs of *Figure 4—figure supplement 1*.

ubiquitin binding protein that would also act as a chaperone. Upon release from this putative ubiquitin-binding chaperone, LRP6 is deubiquitinated by USP19. Following USP19 deubiquitination, LRP6 can, if properly folded, exit the ER, an events possibly promoted by palmitoylation of cysteines in the vicinity of Lys-1403 (*Abrami et al., 2008b*). Or, LRP6 can undergo a new cycle of Lys-1403-specific ubiquitination and chaperone binding. Upon prolonged presence in the ER, LRP6 undergoes polyubiquitination on other cytosolic lysines, leading to ERAD targeting. In the ER, LRP6 can thus undergo two types of ubiquitination events, which operate with different kinetics: (1) specific ubiquitination on Lys-1403 which promotes folding, (2) the slower polyubiquitination on other lysines which promotes ERAD. These two types of ubiquitination events are reminiscent of de-glucosidation and de-mannosidation of glycoproteins in the ER lumen, which also occur with different kinetics. De-glucosidation promotes folding of glycosylated protein, but if folding is too lengthy or fails, de-mannosidation takes place and targets the protein to ERAD.

While the here described ubiquitin-dependent folding system was identified for LRP6, it likely also operates on other membrane proteins. We indeed found that *usp19* silencing also led to a pronounced decrease in the levels of CMG2 (*Figure 4D*), defective folding of which lead to Hyaline Fibromatosis Syndrome (OMIM #228600) (*Deuquet et al., 2009*, *2011*). Silencing of *usp19* however did not affect the levels of the CMG2-related protein TEM8 (Tumor endothelial marker 8), nor those of the Transferrin and the EGF receptors (*Figure 4D*). USP19 was reported to rescue the CFTR Δ508 mutant, which is an ERAD substrate (*Hassink et al., 2009*). USP19 however also controls proteostasis of wild type CFTR (*Figure 4E*). Future studies are required to establish the generality of this ubiquitin-dependent folding system, determine whether it has a built-in quality control, and identify the missing components such as the folding-promoting ubiquitin ligase, the putative ubiquitin-binding chaperone and the degradation-targeting ligase (*Figure 4C*). All these enzymes could be exploited to control Wnt signaling in the context of disease.

# Material and methods

## Cell lines and cell culture

Hela and RPE1 cells were used in this study. These cells are not on the list of commonly misidentified cell lines maintained by the International Cell Line Authentication Committee. They were mycoplasma negative as tested on a trimestral basis using the MycoProbe Mycoplasma Detection Kit CUL001B. RPE1 cells were maintained in DMEM high glucose supplemented with 10% Fetal Calf Serum, 2 mM penicillin and Streptomycin and HeLa cells were maintained in MEM supplemented with 10% FCS, 2 mM L-Glutamine and antibiotics. RPE1 USP19 knockout cell lines were generated using the CRISPR/Cas9 technology. Briefly, the cells were co-transfected with three plasmids: MLML3636, JDS246 carrying the gRNA sequence TCTGGCGGGGCCAGTGCCAC and GFP encoding plasmid. Single GFP transfected cells were sorted by FACS in 96 well plates. The Knockout clones were detected by western blot (*Figure 1—figure supplement 1B* ).

## Antibodies and reagents

We used the following primary antibodies: Rabbit anti-LRP6 mAb (Cell Signaling, #2560 RRID:AB_2139329), Mouse anti-Actin mAb (Millipore, MAB1510), Mouse anti-Myc 9E10 mAb (Covance MMS-

150R RRID:AB_291327), Mouse Ubiquitin (Santa Cruz sc-8017 RRID:AB_628423), Goat anti-CMG2 (R and D systems #AF2940), TEM8 (Sigma-Aldrich, St. Louis, SAB2501028), Mouse anti-Tf-R (Zymed, #13–6800), Mouse anti-EGF-R (Sigma E3138, RRID:AB_476925), CFTR (home-made), Mouse anti-Tubulin (Sigma T5168), Mouse anti-GFP (Roche, #11814460001), Rabbit anti-USP19 (Bethyl, A301-587A, RRID:AB_1078839).

We used the following beads for immunoprecipitations: Protein G Sepharose 4 Fast Flow (GE Healthcare, 17–0618-01), Streptavidin Agarose (Sigma, S1638), anti-Myc Affinity Gel (Thermo Scientific # 20169).

## Plasmids and transfections

LRP6 encoding plasmids as been previously described (*Abrami et al., 2008b*). The catalytically inactive mutant was obtained by Quik Change Technology (Agilent) according to manufacturer's instruction. Plasmids were transfected into RPE1 and HeLa cells for 24 hr in cDNA/9.6 cm$^2$ plate using Fugene (Roche Diagnostics). For the dual Luciferase assay, plasmids and reagents were from Promega.

## Biochemical assays

For immunoprecipitation, cells were PBS washed and lysed 30 min at 4°C in IP Buffer (0.5% Nonidet P-40, 500 mM Tris pH 7.4, 20 mM EDTA, 10 mM NaF, 2 mM benzamidin and protease inhibitor cocktail (Roche), and centrifuged 3–5 min at 5000 rpm. Supernatants were incubated overnight with Sepharose beads. In case of non-coupled Sepharose G beads, supernatants were subjected to pre-clearing with the beads prior to the actual immunoprecipitation reaction.

Cell surface protein biotinylation was performed as described (*Abrami et al., 2008b*). Briefly, silenced cells were treated 30 min with 0.17 mg/ml sulfo-NHS-Biotin (Pierce) at 4°C, quenched with 100 mM NH$_4$Cl, and lysed in IP Buffer (as described above). The lysate was immunoprecipitated with streptavidin-coated sepharose beads (Sigma, S1638).

The dual luciferase assays were performed with plasmids and reagents from Promega. HEK293 cells were transfected with 0.1 µg TOP-Luciferase, 50 ng TK-Renilla, 0.25 µg Mesd encoding plasmid and 0.5 µg myc-LRP6 wild type or mutant encoding plasmids/9.6 cm$^2$ plate. The cells were lysed 24 hr after transfection and the luciferase activity was determined according to the manufacturer's instructions.

The drugs are used in complete medium at the final concentration of 10 µM for MG132 and 100 nM for Bafilomycin A 2 hr before the starvation and are kept during the whole experiment.

## Metabolic and cycloheximide chases

For the metabolic chases, the cells were starved in DMEM HG devoid of Cys/Met for 40 min at 37°C, pulsed with the same medium supplemented with 140 µCi of $^{35}$S Cys/Met for the indicated time, washed and incubated in DMEM complete medium for the indicated time of chase (*Abrami et al., 2008a*).

For the cycloheximide chases, the cells were incubated in medium supplemented with 10 µg/ml of cycloheximide for 2 hr, washed and incubated in complete medium for the indicated time.

## Acknowledgements

We are grateful to Maria Zaballa and Jérôme Bürgi for critical reading of the manuscript and many discussions, to Christophe Fuerer for the constructs to perform Wnt signaling measurements. The work received funding from the European Research Council under the European Union's Seventh Framework Program (FP7/2007–2013) / ERC grant agreement n° 340260 PalmERa and from the Swiss National Science Foundation. MF was the recipient of a long term EMBO fellowship.

## Additional information

### Funding

| Funder | Grant reference number | Author |
|---|---|---|
| European Research Council | 340260 PalmERa | F Gisou van der Goot |
| Schweizerischer Nationalfonds zur Förderung der Wissenschaftlichen Forschung | | F Gisou van der Goot |
| Seventh Framework Programme | FP7/2007-2013 | F Gisou van der Goot |

The funders had no role in study design, data collection and interpretation, or the decision to submit the work for publication.

### Author contributions

EP, LA, MF, Conception and design, Acquisition of data, Analysis and interpretation of data, Drafting or revising the article; BK, Acquisition of data; SU, Drafting or revising the article; FGvdG, Conception and design, Analysis and interpretation of data, Drafting or revising the article

### Author ORCIDs

F Gisou van der Goot, http://orcid.org/0000-0002-8522-274X

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
