## [Decision Letter]

Thank you for submitting your article "Ubiquitin-dependent folding of the Wnt signaling coreceptor LRP6" for consideration by *eLife*. Your article has been favorably evaluated by Ivan Dikic (Senior Editor) and three reviewers, one of whom is a member of our Board of Reviewing Editors. The reviewers have opted to remain anonymous.

The reviewers have discussed the reviews with one another and the Reviewing Editor has drafted this decision to help you prepare a revised submission.

There is a clear consensus between the reviewers that this paper is interesting and describes a significant new form of quality control of membrane protein synthesis and maturation. The data, which are high quality and largely well interpreted, are used to propose a model in which LRP6 is unstable in the ER and degraded by ERAD. Newly synthesized LRP6 is stabilized by ubiquitination of K1403, allowing folding and palmitoylation. Subsequent removal of ubiquitin is necessary for forward trafficking to the Golgi apparatus. Despite the general positivity among reviewers, there are some issues that need to be resolved before it can be accepted.

1) The trafficking route of LRP6 needs further clarification. Given the recently defined misfolded protein-associated secretion pathway termed MAPS from the Ye lab (Lee et al., 2016) that utilizes USP19, does LRP6 trafficking to the cell surface travel through the MAPS pathway that bypasses the Golgi or the traditional secretory pathway?

2) In a related issue, the experiments that show different kinetics of the effects of MG132 and bafilomycin imply that there are two distinct degradation mechanisms. These could be further clarified by examining the endo-H sensitive/resistant LRP fractions that are subject to MG132 or bafilomycin-sensitive degradation. Investigating the glycosylation in this way should also address the above question of whether LRP6 is trafficked conventionally through the Golgi.

3) Late in the paper it is suggested that USP19 might also be the unidentified chaperone that binds to K1403 ubiquitinated LRP6. This is a major mechanistic issue and should be explored a little further. The consequences on Wnt signaling of overexpression of a catalytically dead form of USP19 would be a simple experiment that might separate these functions, allowing the possibility of this dual function to be addressed.

4) There is a significant lack of clarity in the dynamic aspect of the model that needs to be better explained. K1403 ubiquitination is both a stabilizing process for the newly synthesized LRP6 – allowing a chaperone to bind and support folding. But loss of USP19, predicted to increase time or level of Ub binding to K1403, leads to destabilization of LRP6. This appears to contradict the idea that ubiquitination is stabilizing. The resolution of this paradox is probably the timing of two events: first stabilization by ubiquitin addition, then ubiquitin removal for ER exit – but this is not well described, nor is there much data to support it strongly. Do they think that the palmitoylation, which is discussed but not well integrated into the model, is a secondary stabilizing factor that prevents destabilization of mature LRP6 after the ubiquitin is removed? And is there anything to be said about the relationship between K1403 ubiquitination and poly-ubiquitination that promotes ERAD? Please clarify the model generally.

5) Finally, the cell surface expression of LRP6 is measured using a biotinylation assay in Figure 4—figure supplement 1 upon USP19 overexpression. This experiment is missing controls such as LRP6 levels in WCL and a loading control. It would also be of interest to see how overexpressing the USP19 catalytic mutant would affect cell surface expression of LRP6.

---

## [Author Response]

*There is a clear consensus between the reviewers that this paper is interesting and describes a significant new form of quality control of membrane protein synthesis and maturation. The data, which are high quality and largely well interpreted, are used to propose a model in which LRP6 is unstable in the ER and degraded by ERAD. Newly synthesized LRP6 is stabilized by ubiquitination of K1403, allowing folding and palmitoylation. Subsequent removal of ubiquitin is necessary for forward trafficking to the Golgi apparatus. Despite the general positivity among reviewers, there are some issues that need to be resolved before it can be accepted.*

*1) The trafficking route of LRP6 needs further clarification. Given the recently defined misfolded protein-associated secretion pathway termed MAPS from the Ye lab (Lee et al., 2016) that utilizes USP19, does LRP6 trafficking to the cell surface travel through the MAPS pathway that bypasses the Golgi or the traditional secretory pathway?*

We now mention in the Introduction (third paragraph) that upon exit from the ER, LRP6 is transported through the Golgi to the plasma membrane as indicated by the lack of EndoH sensitivity of the mature protein (Abrami et al., JCB 2008). Thus LRP6 follows the traditional secretory route rather than the MAPS pathway.

*2) In a related issue, the experiments that show different kinetics of the effects of MG132 and bafilomycin imply that there are two distinct degradation mechanisms. These could be further clarified by examining the endo-H sensitive/resistant LRP fractions that are subject to MG132 or bafilomycin-sensitive degradation. Investigating the glycosylation in this way should also address the above question of whether LRP6 is trafficked conventionally through the Golgi.*

Figure 1 shows that WT LRP6 degradation is sensitive to both MG132 and Bafilomycin. We now additionally show (new panel Figure 1) that degradation of palmitoylation-deficient LRP6, which does not exit the ER (Abrami 2008), is sensitive only to MG132, not to Bafilomycin. Thus degradation by the Bafilomycin sensitivity pathway requires ER exit. Together these experiments (described in subsection “Massive degradation of newly synthesized LRP6 in the ER”, last paragraph) show that WT LRP6 undergoes degradation via two distinct pathways, ERAD in the ER and, following ER exit, degradation mediated by lysosomes.

*3) Late in the paper it is suggested that USP19 might also be the unidentified chaperone that binds to K1403 ubiquitinated LRP6. This is a major mechanistic issue and should be explored a little further. The consequences on Wnt signaling of overexpression of a catalytically dead form of USP19 would be a simple experiment that might separate these functions, allowing the possibility of this dual function to be addressed.*

We now mention (subsection “USP19 promotes LRP6 biogenesis and controls Wnt signaling”, second paragraph) that USP19 is very *unlikely* to be both the DUB and the ubiquitin-binding chaperone. We show in Figure 3 that over- expression of catalytically inactive USP19 has no protective role towards degradation. Our initial interpretation was limited to the conclusion that the catalytic activity is required for LRP6 rescue, but the complete lack of effect also rules out that USP19 acts as the chaperone since otherwise a partial rescue would have been observed.

*4) There is a significant lack of clarity in the dynamic aspect of the model that needs to be better explained. K1403 ubiquitination is both a stabilizing process for the newly synthesized LRP6 – allowing a chaperone to bind and support folding. But loss of USP19, predicted to increase time or level of Ub binding to K1403, leads to destabilization of LRP6. This appears to contradict the idea that ubiquitination is stabilizing. The resolution of this paradox is probably the timing of two events: first stabilization by ubiquitin addition, then ubiquitin removal for ER exit – but this is not well described, nor is there much data to support it strongly. Do they think that the palmitoylation, which is discussed but not well integrated into the model, is a secondary stabilizing factor that prevents destabilization of mature LRP6 after the ubiquitin is removed? And is there anything to be said about the relationship between K1403 ubiquitination and poly-ubiquitination that promotes ERAD? Please clarify the model generally.*

In the first paragraph of the subsection “Concluding remarks”, we have now clarified that LRP6 can undergo two types of ubiquitination events in the ER and that these events operate with different kinetics: 1) specific ubiquitination on K1403, which promotes folding, 2) the slower polyubiquitination on other lysines, which promotes ERAD. The latter does not have time to take place if the protein folds and exits the ER.

We also mention that, given the proximity of the palmitoylation sites to Lys1403, palmitoylation might prevent re-ubiquitination on Lys-1403, thus preventing re- entry into the cycles of ubiquitination-deubiquitination.

Finally, we mention that the two types of ubiquitination events are reminiscent of de-glucosidation and de-mannosidation of glycoproteins in the ER lumen, which also occur with different kinetics. De-glucosidation promotes folding but if this is too lengthy de-mannosidation takes place and targets the protein to ERAD.

*5) Finally, the cell surface expression of LRP6 is measured using a biotinylation assay in Figure 4—figure supplement 1 upon USP19 overexpression. This experiment is missing controls such as LRP6 levels in WCL and a loading control. It would also be of interest to see how overexpressing the USP19 catalytic mutant would affect cell surface expression of LRP6.*

We thank the reviewers for raising this point. We have now removed this figure, which indeed lacked the appropriate controls. Almost all experiments in the paper were performed by metabolic pulse-chase because, as we mentioned already in the original text, western blot analysis of the steady state population does not allow the analysis of the effect of USP19 by protein overexpression, due to the very high stability of a subpopulation of LRP6, WT and mutants, that end up constituting the steady state population monitored by western blot.

Consistent with this, as we should have realized, surface biotinylation, which is revealed by western blotting, with the requested controls, does not, and cannot, reveal differences of surface LRP6 upon overexpression. Thus, while the Wnt signaling assay, which is more quantitative, indicates that cells overexpressing LRP6, Mesd and USP19 undergo higher Wnt signaling than cells overexpressing only LRP6 and Mesd (Figure 4) (in cells that have endogenous levels of the 3 proteins), this cannot be seen by surface biotinylation. We are happy to add these experiences to the manuscript if requested, however we feel that it might render things unnecessarily confusing for readers.

Instead we have added surface biotinylation to accompany Figure 4, which also illustrates that Wnt signaling is strongly dependent on the USP19 levels, however using USP19 silencing instead of overexpression. As shown in various figures of the paper, the effect of USP19 silencing or knock out on LRP6 levels can readily be observed by western blotting. Consistently surface LRP6, monitored by surface biotinylation, decreases upon USP19 silencing. We show gradual decrease of LRP6 at the cells surface after 1, 2 and 3 days of siRNA. At day 3 of siRNA, LRP6 levels are undetectable at the surface but also very low in the total cell extract. Therefore, quantification of surface LRP6 was performed at day 2 of siRNA, when total LRP6 levels are still detectable (Figure 4—figure supplement 1).